# Exploring Genetic and Neural Risk of Specific Reading Disability within a Nuclear Twin Family Case Study: A Translational Clinical Application

**DOI:** 10.3390/jpm13010156

**Published:** 2023-01-14

**Authors:** Tina Thomas, Griffin Litwin, David J. Francis, Elena L. Grigorenko

**Affiliations:** 1Department of Psychiatry, Massachusetts General Hospital, Boston, MA 02114, USA; 2Texas Institute for Measurement, Evaluation, and Statistics, University of Houston, Houston, TX 77204, USA; 3Hewlett Packard Enterprise Data Science Institute, University of Houston, Houston, TX 77204, USA; 4Department of Psychology, University of Houston, Houston, TX 77204, USA; 5Department of Molecular and Human Genetics, Baylor College of Medicine, Houston, TX 77030, USA; 6Center for Cognitive Sciences, Sirius University of Science and Technology, 354340 Sochi, Russia

**Keywords:** neuroimaging, genetics, reading disability

## Abstract

Imaging and genetic studies have characterized biological risk factors contributing to specific reading disability (SRD). The current study aimed to apply this literature to a family of twins discordant for SRD and an older sibling with reading difficulty. Intraclass correlations were used to understand the similarity of imaging phenotypes between pairs. Reading-related genes and brain region phenotypes, including asymmetry indices representing the relative size of left compared to right hemispheric structures, were descriptively examined. SNPs that corresponded between the SRD siblings and not the typically developing (TD) siblings were in genes *ZNF385D*, *LPHN3*, *CNTNAP2*, *FGF18*, *NOP9*, *CMIP*, *MYO18B*, and *RBFOX2*. Imaging phenotypes were similar among all sibling pairs for grey matter volume and surface area, but cortical thickness in reading-related regions of interest (ROIs) was more similar among the siblings with SRD, followed by the twins, and then the TD twin and older siblings, suggesting cortical thickness may differentiate risk for this family. The siblings with SRD had more symmetry of cortical thickness in the transverse temporal and superior temporal gyri, while the TD sibling had greater rightward asymmetry. The TD sibling had a greater leftward asymmetry of grey matter volume and cortical surface area in the fusiform, supramarginal, and transverse temporal gyrus. This exploratory study demonstrated that reading-related risk factors appeared to correspond with SRD within this family, suggesting that early examination of biological factors may benefit early identification. Future studies may benefit from the use of polygenic risk scores or machine learning to better understand SRD risk.

## 1. Introduction

Specific reading disability (SRD) has both genetic and neural risk factors, characterized by an evolving research literature. However, there has been less focus on the application of this research literature to better understand risk within individuals or families. As our understanding of genetics and neural factors increases, it is likely to contribute to the development of precision medicine, where diagnosis and treatment can be tailored to the individual based on their specific risk factors using methodologies such as polygenic risk scores [1,2]. Reiss and colleagues [3] suggested that genetic studies are beneficial for intervention because they help to identify important target variables for intervention, provide information about mechanisms of effects, may differentiate individual responses to intervention, and help to reveal the best timing for intervention. Genetic risk factors interact with various environmental variables in affecting reading-related outcomes, and this information can be used to help cater treatment to specific individuals [4]. The incorporation of neural factors into the calculation and understanding of risk would further improve the application of this knowledge to help individual families understand diagnoses and intervention options [5], as neural function and structure have been shown to predict and differentiate responses to intervention [6,7,8,9] and change with intervention [10,11,12]. Response to intervention is the gold standard for the diagnosis of SRD, differentiating those individuals who are able to respond given adequate intervention from those who do not [13], and these latter individuals have a different biological risk profile compared to the former.

Phase 1 genetic translational research focuses on the transition from the genome-based discovery of relevant genes to the application of these findings to individuals [14]. The current study examines biological risk factors within an individual family case study, using the SRD imaging genetics literature to select genes and brain structures that may be relevant to risk within the family to determine whether there is evidence of SRD risk within the family. Because the family consists of a pair of twins, one with SRD and one without, as well as an older sibling with reading difficulties, the risk factors that are more similar among those siblings with SRD, are likely important for understanding risk within this family. A recent meta-analysis synthesizing the results of twin studies of SRD suggested that the heritability of reading ability is thought to be 66%, with a shared environment effect of 13%, and a non-shared environment effect of 21% [15]. Given the similar environments of each of these siblings, particularly the twins, the biological factors contributing to the development of SRD may be particularly important in understanding the risk within this family.

Genetic risk factors influence brain structure and function in networks associated with reading development. SRD risk genes have been identified through both candidate gene studies and genome-wide association studies (GWAS). SRD is genetically heterogeneous and is influenced by small effects from many genes. Specifically, much of the genetic research on SRD has been focused on single nucleotide polymorphisms (SNPs), or single substitutions of nucleotides in a genome sequence, which can serve as biological markers or predict genetic risk. Much of the genetic research focused on SRD has been focused on SRD risk loci, named DYX1-DYX9, which reside on eight different autosomal chromosomes in nine chromosomal locations [16]. Within these SRD risk loci, genes such as *DYX1C1*, *DCDC2*, *KIAA0319*, and *ROBO1* have been replicated in most studies. Other genes outside of these specific loci have been identified as well [16,17], including language-related genes *FOXP2* and *CNTNAP2* [18]. Furthermore, a recent genome-wide association study revealed a nominal association of an SNP near the gene *FGF18* with SRD [19]. Results of a linkage analysis suggested relationships between SRD and SNPs within the *MSI2* gene and upstream of the attention-deficit/hyperactivity disorder (ADHD)-related *LPHN3* gene [19]. Similarly, another recent GWAS implicated an uncharacterized gene, *LOC388780*, and the gene *VEPH1*, related to brain development [20].

Incorporating the use of imaging methodology has also improved the understanding of risk factors contributing to SRD. The reading network of the brain has been characterized using various imaging methodologies, most commonly including magnetic resonance imaging (MRI), functional MRI (fMRI), and electroencephalography (EEG), which have been used to characterize brain structures or functions that are related to SRD. Reading-related brain regions include left hemispheric networks, including a dorsal pathway related to phonological processing in the occipital-temporal and the inferior parietal lobe, a second ventral pathway related to automatic word reading in the left temporal lobe and fusiform gyrus, and a frontal network, including Broca’s area, involved in attention and mental verbalization [21]. Both structural and functional studies have shown that this reading network is atypical in individuals with SRD. For example, adults with SRD demonstrated underactivation in superior temporal regions, while children with RD demonstrated underactivation in inferior parietal regions [22]. Similarly, nine studies found reduced grey matter volume in the right superior temporal gyrus and left superior temporal sulcus in SRD [23]. Diffusion tensor imaging studies have indicated that there are lower fractional anisotropy levels in left temporoparietal and frontal regions, including the left arcuate fasciculus and corona radiata, in SRD [24]. Another more recent study demonstrated that higher fractional anisotropy in the right superior longitudinal fasciculus and left inferior cerebellar peduncle were correlated with better nonword reading skills among older children aged 9 and above [25].

There has also been evidence of atypical lateralization or asymmetry in individuals with SRD. For example, children with SRD have been shown to have atypical asymmetry in the inferior frontal-occipital fasciculus (leftward) and superior longitudinal fasciculus (rightward), which in turn is related to reading skills [26]. In turn, children with SRD may have more compensatory activation in right hemispheric regions such as the right superior temporal gyrus [27]. Children without SRD have been shown to have greater rightward asymmetry of the cerebellum compared to those with SRD [28]. These differences in lateralization may exist even before the development of reading, as Guttorm and colleagues [29] showed that pre-reading children under age 5 with a family risk of SRD tended to have atypical speech processing in the right hemisphere on EEG. Similarly, children with no family risk for SRD tended to have greater leftward asymmetry of the planum temporale compared to those with family risk [30]. It has been proposed that reading-related genes such as *DYX1C1*, *ROBO1*, and *DCDC2*, which have functions that contribute to ciliogenesis and cilia function, may contribute to processes contributing to asymmetries such as the development of the corpus callosum or the direction of neuronal migration [31].

Furthermore, research has demonstrated that there are changes in brain structure and function with reading intervention, which can occur both in reading network regions and right hemispheric regions that may help to compensate for weaknesses in the left-hemispheric reading network. For example, studies have revealed increased activity in the left superior temporal gyrus [12] and changes in activation in the left thalamus, right insula/inferior frontal, left inferior frontal, right posterior cingulate, and left middle occipital gyri [10]. Similarly, increases in grey matter volume have been observed following intervention in the left anterior fusiform gyrus/hippocampus, left precuneus, right hippocampus, and right anterior cerebellum [32].

A greater understanding of these genetic and imaging factors may hold promise for using these methodologies for the clinical assessment of disorders such as SRD. Within clinical work, in the assessment of dementias, brain injuries, or other illnesses that impact the brain, the current standards for detecting brain abnormalities are visual inspection by experts. However, research has shown that clinical interpretation based on visual inspection can be unreliable [33,34]. The use of unbiased quantitative tools may improve the detection of brain-related disorders, even for individuals who are minimally trained [34]. For example, Hedderich and colleagues [35] demonstrated that the use of normative brain volume reports, comparing volumes of brain structures with a healthy sample, can improve the identification of patients compared to healthy controls and improve interrater reliability. While imaging tools using MRI scans to assess disorders such as Alzheimer’s and multiple sclerosis have been developed, there has been less application to other types of learning or psychiatric disorders, potentially because of the smaller effects of many genes and structures, a lack of established biomarkers, and overlap across various learning and psychiatric disorders [36]. Development of quantitative tools would be particularly relevant for SRD because brain differences are not detectable by visual inspection, but there may be patterns (e.g., structural or functional characteristics of individual regions of interest (ROIs) such as cortical thickness, cortical surface area, volume, or brain asymmetry) that may be important for detecting the early risk of SRD. Furthermore, using novel methodologies and approaches to quantify characteristics of brain structures or networks at the individual level may improve or contribute to the development of quantitative tools to assess SRD. 

Similarly, as the understanding of genetic risk factors for SRD increases, research may focus on using additional methodologies for the application of genetic testing and intervention, incorporating genetics into evidence-based guidelines and health practice, and evaluating the beneficial clinical impact of this type of research [14]. However, for SRD, there is often variability across genome-wide association studies, and there are likely genetic factors that have not yet been identified. There may also be significant interaction with environmental factors in affecting risk [37]. The use of a polygenic risk score to predict SRD revealed significant overlap in etiology with word reading and other neurodevelopmental disorders such as ADHD and autism spectrum disorder (ASD), as well as general cognitive ability [38]. While there is no current diagnostic tool based on biological factors, the aim of the current study is to apply the current literature on imaging and genetic factors contributing to SRD to better understand the specific genetic and imaging risk factors that may be related to the development of SRD in an individual family, including a pair of twins, one with reading difficulty and one without, and an older sibling with reading difficulty as well. The current study serves as an exploration of impact within the individual family as part of phase 1 translational research, using both existing genetic and imaging phenotypes (e.g., cortical thickness) and also using a novel approach to examining brain images in the individual. The unique profile of this family allowed examination of how genetic and imaging patterns, including structural asymmetry, covary with SRD. Specifically, patterns were descriptively examined to identify relevant brain region phenotypes (including cortical thickness, cortical surface area, grey matter volume, and asymmetry) and single nucleotide polymorphisms that were more similar among the siblings with SRD, as compared to the twins. This also allowed us to determine which specific genes and structures, within the larger group of known reading-related genes and brain structures, were most related to SRD within this sample. Furthermore, a new image analysis technique was developed and tested using images derived from structural magnetic resonance imaging (MRI) data. This technique presents an efficient method of evaluating physical characteristics related to the structure of selected regions, using a novel methodology to understand risk within a small case sample.

## 2. Methods

### 2.1. Participants

Participants were members of a White/Caucasian family, consisting of a pair of dizygotic twins (age 9), one with a diagnosed SRD and one without, an older sibling (age 11) with a reported history of reading difficulty, and their parents. All siblings were female. Imaging and behavioral data were collected from the three siblings, and genetic data were collected from all five family members. The collection of this data was approved by the University of Houston Institutional Review Board (IRB# HSC-MS-12-0259 and CR00001300). Written informed consent was obtained from the parent or legal guardian of minor participants.

### 2.2. Behavioral Assessments

Assessments of word reading included the Test of Word Reading Efficiency (TOWRE; [39]), a timed measure of an ability to read printed words, the Kaufman Test of Educational Achievement, 3rd edition (KTEA-3; [40]), including the subtests Letter Word Recognition, measuring word reading ability, Nonsense Word Decoding, measuring decoding abilities, Reading Comprehension, and Listening Comprehension.

### 2.3. Genetic Data

Oragene saliva kits (DNA Genotek, Inc., Kanata, ON, Canada) were used to obtain saliva samples during behavioral testing sessions. Genomic DNA was extracted from the samples using the FlexiGene DNA Kit (Qiagen, Hilden, Germany) per the manufacturer’s protocol. Genotyping was carried out at the Human Genome Sequencing Center of the Baylor College of Medicine according to the manufacturer’s instructions (Illumina, San Diego, CA, USA). Genotyping was completed with the Infinium CoreExome-24 v1.4 BeadChip, which contains 567,218 probes. Illumina’s GenCall algorithm was run on the raw genetic data to cluster and call genotypes and assign confidence scores. QC filtering was applied to each sample separately, with a no-call threshold of 0.15. Therefore, all genotypes with a GenCall score less than or equal to 0.15 were assigned as missing, since they are considered too far from the cluster centroid to be reliably genotyped. Variants for 28 genes of interest, reading-related genes, were extracted to examine whether genotypes varied with reading difficulty within the family.

### 2.4. Imaging Data

MRI data were collected to obtain information on brain morphometry. The data were collected at the Core for Advanced Magnetic Resonance Imaging (CAMRI) at Baylor College of Medicine. Structural data covering the whole brain was obtained using a Siemens 3T scanner with a 64-channel head coil and slice-accelerated, simultaneous multislice imaging sequence (0.8 mm^3^ slice thickness, FOV = 256 × 256, TR/TE = 2400/2.22 ms, α = 8 [7]. Freesurfer v6.0.0 software [41,42] was used to obtain high-resolution anatomical images with an accelerated 3dT1-weighted sequence [7]. Imaging data was registered to “fsaverage,” a Freesurfer template brain based on the average of many MRI scans, to allow comparisons across participants. ROIs were defined a priori. ROIs were first identified and automatically masked (annotation layers indicating the ROIs) using the Destrieux atlas [43], and the resultant masks were manually modified and fitted to ensure that the labeled ROIs only included grey matter and not white matter. Freesurfer was used to generate statistics, including grey matter volume, cortical thickness, and cortical surface area, selected to better understand the underlying grey matter factors that may be related to reading disability within a clinical case study. Heritability tends to be higher for cortical surface area and volume, while thickness is more likely to be impacted by environmental factors [44], helping to differentiate phenotypes that may represent underlying risk factors, and those that may be related to compensation.

Additionally, we conducted novel analyses with MRI data using image analysis techniques to explore the relationship between ROI morphology and SRD risk within the family. This approach involved a pairwise comparison of ROI mask pixel luminosity distributions between siblings. MRI data are processed to graphically represent physical structure as pixels in MRI image “slices” along each of the three anatomical planes. Grayscale two-dimensional image formats encode visual information as a matrix of “channel values,” with the number 0 corresponding to black and 255 to white, with intermediate values as shades of gray.

For each of our eleven identified ROI (inferior occipital; fusiform gyrus; inferior frontal gyrus, pars orbitalis; inferior frontal gyrus, pars triangularis; inferior frontal gyrus, pars opercularis; supramarginal gyrus; angular gyrus; superior temporal gyrus; transverse temporal gyrus; planum temporale; planum polare), each MRI was masked as described above. These three-dimensional masks were exported as individual two-dimensional slices into PNG image format, yielding approximately 30–70 images per axis (sagittal, coronal, and axial) of an ROI, and a total of more than 4000 images. Composite images were compiled for each anatomical plane of each region using the Python computer-vision libraries opencv and PIL [45,46]. Each composite was programmatically created by adjusting the alpha (opacity) channel of each image and loading a set of ROI mask slices into one multi-layered image file. This use of opacity modification and layering results in an image in which brighter areas indicate “thicker” or higher volume cortical areas, because those pixels were present in more slices. The number of slice layers present in a particular mask is influenced by total brain size and patient position within the scanner. To account for this, the alpha adjustment was standardized for each region between subjects.

The resulting images (Figure 1) show a heatmap of volume (area defined by x and y pixel positions, with the z dimension represented by luminosity). A distribution of luminosity values was extracted from each image using the R statistical software with image-processing packages [47,48,49], and these distributions were the basis for constructing Q–Q plots to evaluate the pairwise degree of dissimilarity between patients for each ROI.

### 2.5. Analyses

For genetic analyses, reading-related risk genes were identified from the literature, and included *MRPL1*9, *ZNF385D*, *ROBO1*, *VEPH1*, *LPHN3*, *FGF18*, *DCDC2*, *KIAA0319*, *TTRAP*, *THEM*2, *RIPOR2*, *CMAHP*, *FOXP2*, *CCDC136*, *CNTNAP2*, *SLC2A3*, *COL4A2*, *NOP9*, *TUBGCP5*, *CYFIP1*, *NIPA2*, *SEMA6D*, *DNAAF4*, *CMIP*, *ATP2C2*, *MSI2*, *MYO18B*, and *RBFOX2*. Genotypes of SNPs associated with these previously identified reading-related genes were determined. SNPs that were the same among all family members were removed, leaving 684 SNPs remaining for the analysis. Genetic patterns were examined to identify SNPs for which the genotypes were the same between the sibling with reading difficulties, but for which the twin differs. The proportion of genotypes following this pattern was compared between SRD risk genes and the rest of the genome.

For imaging analyses, multiple phenotypes were used to examine patterns in cortical thickness, surface area, and grey matter volume in the reading network. Pairwise intraclass correlations (ICCs) were used to determine the degree of similarity of imaging phenotypes between each pair of twins, allowing us to determine how closely the siblings resemble each other. The ICCs were calculated for each phenotype (e.g., cortical thickness) using average values for each region of interest covering the whole cortex, and then for just the reading-related regions (fusiform gyrus, inferior parietal, banks superior temporal sulcus, inferior frontal gyrus pars opercularis and pars triangularis, supramarginal gyrus, and transverse temporal gyrus).

As a measure of lateralization of regions, asymmetry indices were calculated for phenotypes in specific regions of interest. Asymmetry indices were calculated as (L − R)/((L + R)/2). A positive value indicates leftward asymmetry, while a negative value indicates rightward asymmetry. Imaging phenotypes and asymmetry were plotted to visualize patterns between the siblings and identify regions where phenotypes may be related to SRD.

Due to the fundamentally comparative nature of this enquiry—these distributions are meaningless when considered in isolation—analysis focused on two types of quantile-quantile (Q–Q) plots constructed from these distributions. The first construction provides direct inter-patient comparison, in which ROI distributions for patients were arranged along X and Y axes in a pairwise fashion. With three patients, this resulted in 3 Q–Q plots per ROI showing each of the three pairs: the “twin-twin” pairing, the “SRD” pair, and lastly, a pair consisting of the typically developing (TD) twin and the SRD affected older sibling. The “twin-twin” pair presents a baseline, which is expected to show the most similarity [50,51]. A pathological process may be implied by distributions that show increased similarity of the “SRD” pair in that region (for regions in which the development of an area is correlated with SRD), or by distributions that show similarity between the TD twin and the SRD older sibling (especially for regions that are thought to develop in a continuous fashion with a faculty such as reading skill). Next, intra-patient comparisons were constructed in which pairs of ROIs were arranged in Q–Q plots; pairs were constructed as all combinations of member ROIs of the frontal, ventral, and dorsal pathways (Table 1).

Comparisons within the first group allow for the evaluation of relative dissimilarity in structure between siblings; divergence provides evidence of dissimilar structure. The second group allows for a more subtle distinction: first, differences in ROI-pair distributions allow for a baseline comparison of expected variation between regions, which contextualizes the relationships observed in the first group; second, relative regional dissimilarities shared between the reading-difficulty siblings further implicate specific brain structures that may be relevant to reading disorder.

## 3. Results

Behavioral data (see Table 2) revealed low average reading abilities for one twin and the older sibling, and average reading abilities for the second twin based on the age norms of the tests. Scores tended to be lower in those siblings with SRD in both word reading and decoding, as well as reading comprehension.

### 3.1. Genetic Analysis

In the genetic analysis, there were 79 SNPs where the siblings with SRD had the same genotype, but the typically developing twin did not. This was about 10% of the overall SNPs within the “reading” genes (i.e., genes, previously associated with reading componential phenotypes) examined, consistent with the rest of the genome. Overall, there were 33 SNPs in *ZNF385D*, 17 SNPs in *LPHN3*, 9 SNPs in *CNTNAP2*, 2 SNPs in *FGF18*, 2 SNPs in *NOP9*, 11 SNPs in *CMIP*, 4 SNPs in *MYO18B*, and 1 SNP in *RBFOX2.* Fifty-seven of these SNPs were intronic, 1 SNP was exonic (in *NOP9*), 1 SNP was in the three prime untranslated regions, and 20 SNPs were intergenic.

### 3.2. Cortical Thickness

Pairwise ICCs were used to determine the degree of similarity of average cortical thickness across regions of interest across the whole brain and then across just reading-related regions. For cortical thickness in regions of interest across the whole brain, the ICCs were similar across each pair of siblings (0.85 for the twins, 0.87 for the siblings with SRD, and 0.82 for the last pair of siblings), and all were significant. Across reading-related regions, the ICC for cortical thickness was highest for the siblings with SRD (ICC = 0.76). The ICC was 0.66 for cortical thickness in reading-related ROIs in the twins, and the ICC was 0.54 for the TD twin and older sibling.

For asymmetry indices calculated for cortical thickness, regions that showed greater correspondence between the two siblings with reading difficulty, as compared to the twins, were determined (see Table 3 and Figure 2). In the transverse temporal gyrus, siblings with SRD showed symmetric or slightly leftward asymmetry, while the twin without SRD had greater thickness in the right hemispheric transverse temporal gyrus. There was a similar pattern, with greater symmetry in the impacted siblings and greater thickness in the right hemisphere in the typically developing (TD) sibling in the superior temporal gyrus. These regions were examined in more detail through the creation of plots of average cortical thickness values in these regions.

When average cortical thickness values were examined in these regions (Figure 3), the TD sibling appeared to have lower cortical thickness in the left hemisphere as compared to the SRD siblings, and higher cortical thickness in the right hemispheric transverse temporal gyrus as compared to the SRD siblings. In the superior temporal gyrus, siblings with SRD tended to have greater cortical thickness in the left hemisphere as compared to the TD sibling (Figure 3). In the right hemisphere, the SRD twin had the highest cortical thickness, followed by the TD twin, and then the older SRD sibling (Figure 3). Therefore, the left hemisphere may have been a greater driver of the asymmetry pattern in the superior temporal gyrus.

### 3.3. Grey Matter Volume

Pairwise ICCs were used to determine the degree of similarity of grey matter volume across regions of interest across the whole brain, and then across just reading-related regions.

ICCs were similar across each pair when examining the grey matter volume of regions, with high levels of correspondence, and all were statistically significant. For grey matter volume in regions across the whole brain, the ICC was 0.97 for the twins, 0.98 for the SRD siblings, and 0.97 for the TD twin and older siblings. Across reading-related regions, the ICCs were similar, with ICCs of 0.97 for the twins, 0.99 for the SRD siblings, and 0.95 for the TD sibling and older sibling. Reading-related regions where volume asymmetry was more similar between SRD siblings as compared to the twins included the fusiform gyrus and the supramarginal gyrus (Table 3). For both of these regions, the TD sibling had greater leftward asymmetry, indicating greater volume in the left hemisphere.

In the fusiform gyrus, the SRD twin appeared to have the highest volume in both the right and left hemispheres (Figure 4). When examining global grey matter volume in these regions, the SRD siblings do not have more similar grey matter volumes in each hemisphere. However, both the SRD siblings have a higher volume in the right hemisphere as compared to the left, while the TD sibling has a higher volume in the left hemisphere.

In the supramarginal gyrus, the SRD siblings had a higher volume than the TD siblings in both the left hemisphere and right hemisphere, with greater overall differences in the right hemisphere (Figure 4).

### 3.4. Cortical Surface Area

Pairwise ICCs were used to determine the degree of similarity of cortical surface area across regions of interest across the whole brain, and then across just reading-related regions. ICCs for the cortical surface area were similar across each sibling pair, and all were statistically significant. For the cortical surface area in regions of interest across the whole brain, the ICC was 0.97 for the twins, 0.99 for the SRD siblings, and 0.96 for the TD sibling and older sibling. For the cortical surface area in reading-related regions of interest, the ICC was 0.96 for the twins, 0.99 for the SRD siblings, and 0.95 for the TD sibling and older sibling.

Reading-related regions where surface area asymmetry was more similar between SRD siblings as compared to the twins include the fusiform gyrus, supramarginal gyrus, and transverse temporal gyrus (Table 3). For all of these regions, the TD sibling had a more leftward asymmetry, indicating a greater cortical surface area in the left hemisphere as compared to the right.

In a comparison of the global cortical surface area in each region (Figure 5), the surface area of the fusiform gyrus was observed to be higher for the SRD twin in both hemispheres. In the left hemisphere, the TD twin and SRD older sibling had a more similar cortical surface area. In the right hemisphere, the SRD twin had the highest cortical surface area, the older sibling with SRD had the next highest area, and the TD twin had the lowest surface area. Both siblings with SRD appeared to have greater differences in surface area between hemispheres, with a higher surface area in the right hemisphere.

In the supramarginal gyrus, the TD sibling had a lower cortical surface area than the SRD siblings in both hemispheres. The magnitude of difference was observed to be greater in the right hemisphere. For the transverse temporal gyrus, the TD sibling had a lower cortical surface in both hemispheres as well.

### 3.5. Image Analyses

#### 3.5.1. Ventral Pathway

Q–Q plots were observed to better understand the similarity between luminosity, reflective of shared structure, between each sibling pair (labeled on the X and Y axes) for each brain ROI. Observation of Q–Q plots of brain regions that were part of the ventral pathway of the brain, related to automatic recognition of words in reading, revealed that in the fusiform gyrus, the typically developing twin and older sibling were the most similar in their luminosity distributions, suggestive of more similarities in structure between the TD twin and older sibling (Figure 6). In comparison, in the inferior occipital region, the twins were more similar in the distribution of luminosity.

Examination of ROI comparison plots (Figure 7), determining the correspondence between the fusiform and inferior occipital gyrus in each sibling, suggests that the relationship between the two structures is more similar for the typically developing twin and older sibling as compared to the sibling with reading disability.

#### 3.5.2. Dorsal Pathway

In Q–Q plots of ROIs in the dorsal temporo-parietal pathway, involved in phonological processing in reading, most regions had the greatest similarity between the twins (including the supramarginal gyrus, the planum temporale, and the transverse temporal gyrus). However, for the superior temporal region, the siblings with reading disability tended to have more similar distributions of luminosity values (Figure 8). 

In the plots comparing regions of interest within each individual, for many of the comparisons, the relationships between structures were similar for each of the twins (the supmarginal/angular, supramarginal/transverse temporal, supramarginal/planum temporale, angular/superior temporal, angular/planum temporale, angular/planum polare, superior temporal/planum temporale, superior temporal/planum polare, transverse temporal/planum temporale, transverse temporal/planum polare, planum temporale/planum polare pairs). For the superior temporal/transverse temporal comparison, superior temporal/supramarginal, and angular/transverse temporal comparison, the siblings with reading disability tended to have more similarity between structures (Figure 9).

#### 3.5.3. Frontal Regions

In Q–Q plots of ROIs in the inferior frontal gyrus, most regions tended to be similar between sibling pairs (Figure 10).

In the plots comparing regions of interest within each individual, for many of the comparisons, the relationships between structures were similar for each of the twins (in the supmarginal/angular, supramarginal/transverse temporal, supramarginal/planum temporale, angular/superior temporal, angular/planum temporale, angular/planum polare, superior temporal/planum temporale, superior temporal/planum polare, transverse temporal/planum temporale, transverse temporal/planum polare, planum temporale/planum polare pairs). For the superior temporal/transverse temporal comparison, superior temporal/supramarginal, and angular/transverse temporal comparison, the siblings with reading disability tended to be more similar (Figure 11).

For ROI comparisons within an individual, the distribution comparisons again appeared somewhat similar between all the siblings for the pars opercularis/triangularis comparison. There was more similarity between the typically developing twin and older sibling in the ROI comparisons between the pars orbitalis/pars triangularis and pars orbitalis/pars opercularis.

## 4. Discussion

The current study used a family case to better understand the translation of research-based findings to application in clinical practice. Whereas many studies have helped identify the brain structures comprising the reading network and the genetic risk factors contributing to the development of SRD at the group level, little research has been conducted on how these results may translate to an individual family or a single participant. To achieve this translational goal, intraclass correlations were calculated to understand the similarity of brain phenotypes between each sibling pair and how neural risk factors are related to SRD. In addition, the variability in the reading-related genes and brain structures that were more similar among siblings with SRD, as compared to twins with differing reading abilities, was descriptively examined by comparison of genotypes and visualization of imaging phenotypes to determine which regions were most relevant to SRD within this family. Finally, a novel methodology was used to examine the distribution of luminosity, or the thickness of grey matter in a single plane, between the three siblings. A summary of the primary and most relevant results is included in Table 4.

### 4.1. Genetic Findings

Overall, there were many SNPs that were identical (shared) among the SRD siblings, consistent with expectations based on the current genetics literature. Relevant SNPs within reading-related genes were first identified by examining which genotypes were more similar between the SRD siblings as compared to the twins, of the reading-related genes that were identified in the literature. The gene with the most SNPs fitting this pattern was *ZNF385D*, which has been previously associated with SRD as well as overall fiber tract volumes and global brain volume [52]. *ZNF385D*’s functions are in nucleic acid binding [53]. *LPHN3* and *FGF18*, identified to be related to SRD from a GWAS [19], also had SNPs following the selected pattern. *LPHN3*, previously associated with ADHD [54], is involved in cell adhesion and signal transduction [53]. *FGF18* codes for a fibroblast growth factor, involved in processes such as mitogenesis, cell proliferation, cell differentiation, and cell migration, and has been shown to be related to the development of cerebellar structures in mice [53].

Another gene with SNPs fitting the expected pattern was *CNTNAP2,* which codes for a member of the neurexin family and functions in cell adhesion [53]. *CNTNAP2* has been shown to be related to nonword reading and language abilities [55]. In addition to SRD, it has also been associated with autism spectrum disorder, ASD, intellectual disability, ID, language impairment, and schizophrenia [56] In a Chinese sample, the relationship between *CNTNAP2* and risk of SRD was higher in females [57]. *NOP9*, which had 2 SNPs following the chosen pattern, has been associated with language through the effects of paternal SNPs on child reading [58,59]. *NOP9* codes for a binding protein that may regulate cellular processes such as transcription and translation [53].

*CMIP*, which contained 11 SNPs varying with SRD, has been related to individual differences in reading skills [60], as well as short-term memory skills in language impairment [61]. It has also been weakly related to SRD in a Chinese sample [62]. *CMIP* is involved in T-cell signaling [53]. Four SNPs in *MYO18B*, which has been associated with math skills in children with SRD [63], fit the relevant pattern as well. *MYO18B* has functions in nucleotide binding, intracellular trafficking, and motor activity [53]. There was one SNP in *RBFOX2* that varied with SRD in the siblings. *RBFOX2*, which is a regulator of alternative splicing in neurons [53], has been associated with reading and language in a genome-wide association study [64]. Further imaging genetic investigations have revealed associations with cortical thickness, particularly in the left parahippocampal gyrus, right middle temporal gyrus, right inferior frontal gyrus, and bilateral superior temporal gyrus [65].

The described SNPs in reading-related genes were all shown to be more similar between the siblings with SRD as compared to the twins, suggesting that these specific SNPs may be important for understanding reading-related risk within this family. However, a limitation is that there may be SNPs fitting this pattern that are not related to reading, and additional SNPs that were not investigated in this study. Therefore, while this exploratory analysis identified SNPs that may be important for understanding SRD in this particular family, future research should further investigate these SNPs using statistical analyses, specifically examining their relationship to reading measures and brain structures. Furthermore, future research may use a polygenic risk score to better understand the cumulative and interacting effects of all of these identified SNPs in conveying the risk of SRD. Additionally, with the developing genetics methodology and the widening field, there have been more genes identified as related to SRD in GWAS studies, providing more targets for future research.

### 4.2. Neuroimaging Findings

Imaging results, when comparing the overall similarity between sibling pairs using ICCs, suggested that measures of grey matter volume and cortical surface area were highly similar among all three siblings. There was more variability when examining the similarity of cortical thickness between sibling pairs. The ICCs for cortical thickness between sibling pairs ranged from 0.82–0.87 when calculated across regions covering the whole cortex. However, when the focus was on cortical thickness in only reading-related regions, the siblings with SRD tended to have a greater degree of similarity as compared to the twins, with the TD twin and older sibling having the lowest similarity. While this is a small sample and it is impossible to determine whether the ICCs are significantly different from each other, they provide a descriptive understanding of the range of values of similarity, particularly with distinctions for cortical thickness. This suggests that cortical thickness may be a key differentiator of SRD risk within individual families, while measures of grey matter volume and cortical surface area may be less sensitive.

Visualization of individual brain regions revealed differences in cortical asymmetry of cortical thickness in the transverse temporal gyrus and superior temporal gyrus. The siblings with SRD were observed to have more cortical symmetry, while the typically developing siblings had greater rightward asymmetry of cortical thickness. Regarding volume, the typically developing sibling tended to have greater leftward asymmetry of the fusiform gyrus and supramarginal gyrus. There was a similar pattern with cortical surface area, as the typically developing sibling had greater leftward asymmetry of cortical surface area in the fusiform gyrus, supramarginal gyrus, and transverse temporal gyrus as compared to the siblings with SRD. Overall, the typically developing sibling had more leftward asymmetry of grey matter volume and cortical surface area in reading-related regions, and more rightward asymmetry of cortical thickness. While typically, a reduced leftward asymmetry would be expected in SRD [26], the greater rightward asymmetry of cortical thickness in the typically developing sibling may reflect compensation, as cortical thickness tends to be more impacted by environmental factors [44]. Grey matter volume and cortical surface area, conversely, fit the pattern of having a greater leftward asymmetry in the typically developing sibling, which may be more reflective of the genetic risk of SRD given the higher genetic association of these phenotypes [44].

These results also corresponded with previous literature investigating the asymmetry of cortical structures in relation to SRD. For example, a prior study found that children who had no family risk of SRD had greater leftward asymmetry of cortical surface area in the planum temporale, in the superior temporal region posterior to Heschl’s gyrus, as compared to those with risk [30]. Leonard and colleagues [66] created an anatomical risk index based on leftward asymmetry of the planum temporale, combined plana, the cerebellar anterior lobe, rightward asymmetry of cerebral volume, and larger overall values for cerebral volume and surface areas of Heschl’s gyri. They found that negative risk indices, indicated by smaller and more symmetrical brain structures, were related to comprehension deficits, while positive risk indices, or having larger, asymmetrical brain structures, were related to poor word reading [66,67,68]. In our sample, the superior temporal region was characterized by greater symmetry in SRD siblings, which is typically more related to comprehension deficits according to Leonard and colleagues’ [66] anatomical risk index. However, the siblings with SRD in our sample had difficulties with both reading comprehension and word reading. Atypical asymmetry has been shown in relevant white matter structures as well, as children with SRD were shown to have reduced leftward lateralization of white matter structure in the inferior frontal-occipital fasciculus, and greater rightward asymmetry of the superior longitudinal fasciculus [26], which could be a potential target of future research. 

Identified reading-related regions of interest that were fitting the selected pattern included the fusiform gyrus, supramarginal gyrus, transverse temporal gyrus, and superior temporal gyrus. Previous research has demonstrated that an increased size of Heschl’s gyrus has been previously associated with decreased reading performance [69], and the overall size has been used to distinguish children with difficulties in phonological decoding as compared to overall verbal ability [70]. Reduced cortical thickness was found in the left Heschl’s gyrus in children who later developed SRD [71], while cortical thickness in the left superior temporal cortex is positively associated with word reading [72].

For the fusiform gyrus involved in visual word recognition [73], greater size has been associated with better reading skills [74], and there have been findings of reduced grey matter in both the left and right fusiform gyrus in children with SRD [75], indicating that there may be effects in both hemispheres, although larger volumes tend to be related to better reading. In addition, following intervention, children with SRD demonstrated increases in grey matter volume in the left anterior fusiform gyrus [32]. Furthermore, previous literature demonstrates positive associations between grey matter volume in the left supramarginal gyrus and reading [76] and smaller gray matter volume in the right supramarginal gyrus in children with SRD relative to TD children [75].

In the current study, measures of global cortical thickness, volume, and surface area within regions of interest conflicted with previous literature, but patterns of asymmetry better corresponded with the expected results. Within this clinical case study, it was harder to interpret the global sizes of relevant brain regions without the context of comparison; therefore, the use of measures such as asymmetry may be more relevant for understanding reading risk within an individual. Using measures such as asymmetry to help consider sizes or characteristics of structures relative to other structures in an individual’s brain, improves understanding of relationships between various brain structures. A comparison of the relative sizes of structures may be more meaningful in understanding an individual’s SRD risk. Furthermore, the relative change in structure in an individual may be an important indicator of reading development, particularly if these changes are compared relative to other brain regions.

Results of the image analysis indicated that for the inferior frontal gyrus and the majority of the ventral pathway, the luminosity of the regions of interest tended to be more similar for the twins, which is what would be expected based on their age and shared environment. However, the distribution of luminosity was more similar for the siblings with reading disability in the broad superior temporal gyrus, in the dorsal pathway, suggesting that this could be a region that could be related to vulnerability to SRD, consistent with the literature [72]. This is also consistent with the asymmetry analyses conducted as part of the current study, as the siblings with SRD tended to have more similar patterns of asymmetry in regions of the dorsal pathway, including the supramarginal gyrus and superior temporal. In comparison, in the ventral pathway, the typically developing twin and older sibling had more similar distributions. Because the older sibling with SRD and the typically developing twin may have more similar reading skill levels, the fusiform gyrus may be a better indicator of overall reading skill development. This is also consistent with the function of the area being related to automatic recognition of words, which develops as reading skills improve. Furthermore, grey matter volume in the fusiform gyrus has been shown to increase following reading intervention in dyslexic children [32]. Consistently, the typically developing sibling and older sibling with SRD had more similar grey matter volumes of the fusiform gyrus in the left hemisphere, and it was only when examining the asymmetry between the left and right hemisphere that the SRD siblings were corresponding, suggesting that examination of structures in both a single hemisphere and in both hemispheres may be helpful for understanding the pattern of risk.

These distributions of luminosity represent the thickness or depth in each of the three dimensions or planes. While not a replacement for grey matter volume or cortical thickness, examining the spatial distribution of luminosity provides an additional measure that may be informative about shape, as well as a heatmap of how thickness in a single plane varies across the regions of interest. Future studies may benefit from including more measures of shape, particularly in a larger sample, to better understand how luminosity may correspond with typical measures of brain structure. Future studies could also examine the asymmetry of luminosity between the same structure on both sides of the brain. Further validation, such as a determination of the baseline of this measure in healthy individuals, would also provide valuable context.

## 5. Conclusions

The current study used measures of asymmetry as well as global measures of grey matter volume, cortical thickness, and cortical surface area, and a novel measure of the distribution of luminosity across three anatomical planes to examine the clinical application of research findings to better understand risk factors related to SRD within a single family, a promising start to phase 1 of genetic translational research [14]. The application of various techniques to the case family allowed the descriptive examination of multiple contributing neural and genetic risk factors for SRD within the family. Using multiple methodologies allowed for an investigation of established measures of brain morphometry (e.g., grey matter volume, cortical thickness, and cortical surface area) and genetics, as well as an examination of the impact of more novel or infrequently studied measures (e.g., luminosity and asymmetry). Using these multiple methodologies allowed examination of the potential risk suggested uniquely by each measure, but also the relationships between them and determination of the factors that may be most relevant for understanding risk. It is likely that the genetic and neuroimaging signatures of reading disability are interlinked, but this was not able to be directly investigated in the current study. Reading-related genes, such as those involved in cilia function, may affect brain development and asymmetry through their effect on processes such as neuronal migration [31], which could affect these structures even before reading develops. For example, children with a family risk of SRD show atypical lateralized speech processing measured through EEG, which is further related to reading [29].

Limitations of the current methodologies include the inherent difficulties of conducting a case study with a limited number of participants; while descriptive examinations of relevant variables could be studied, it is impossible to determine causality without a larger sample to conduct direct analyses. While these studies demonstrate the link between the asymmetry of brain structures and reading, it is difficult to determine the directionality or causality of these effects, whether asymmetry influences reading or vice versa, or whether this interaction may change over time [77]. Furthermore, SRD is complex and can be impacted by a multitude of factors, in addition to neural and genetic risk factors. Even children who may be at genetic risk for developing SRD do not always develop SRD. Similarly, even if a child is not at genetic risk, they may develop SRD due to other risk factors. Understanding more about biological risk factors simply serves as a starting point to identify children who may benefit from further monitoring or intervention.

The findings demonstrated that some of the critical brain regions and genes that have been previously associated with SRD are also co-varied with SRD in the current family, including the transverse temporal gyrus, supramarginal gyrus, and fusiform gyrus. In addition, cortical thickness in reading-related regions tended to be overall more similar among the SRD siblings as compared to the twins, demonstrating that this may be a critical variable in understanding risk. Examination of the distribution of luminosity in each ROI across each anatomical plane revealed that regions in the dorsal pathway tended to be more similar in the twins or siblings with SRD, while regions in the ventral pathway tended to be more similar among the typically developing and older sibling, suggesting that while regions involved in phonological processing could be more related to risk of SRD, regions involved in automatic word recognition may be more related to reading skill level. Genotype patterns were also used to understand which specific reading-related SNPs may be related to SRD in the family as well. While examining the genetic and neural factors that co-segregate with SRD can provide us with descriptive information about potentially relevant risk factors, it is impossible to establish causality with such a small sample. However, given that the brain regions and genes that covaried with SRD were generally consistent with the previous literature, the current study demonstrates the promise of using biological risk factors to better identify risk. Some more recent studies have suggested some inconsistencies in the neural correlates of SRD, suggesting that there may be diversity in factors impacting reading abilities [25,78]. As the literature develops to better understand this variability, the use of refined methodologies such as machine learning may better help to predict SRD within individuals. Furthermore, as knowledge of the genetic factors contributing to certain disorders increases, the idea of “precision medicine” and tailoring interventions based on an individual’s specific risk factors has been developing [2,3,4]. Earlier identification of these risk factors in clinical cases may help clinicians and families understand who may benefit from early reading intervention, allowing earlier access to services. In addition, as the response to intervention research increases, it may be helpful to specifically investigate the biological factors that differentiate those that respond compared to those that do not respond to reading intervention through an understanding of gene–environment interactions. For example, SNPs close to the *DRD2* gene were associated with a level of improvement from a working memory intervention [79]. Similarly, various imaging phenotypes have been shown to predict or differentiate those who respond to reading interventions [6,7,8,9]. Using response to intervention in research on biological factors may help to identify those genes that better differentiate those individuals with a true SRD, which can further help with the diagnosis and tailoring of individualized interventions [2]. Furthermore, biological risk factors may help us to better understand the population of individuals with SRD, including subgroups, as there is significant variability in the phenotype of SRD as well as overlap with other neurodevelopmental disorders [38].

Neural factors, including structure and function, are also highly relevant for understanding risk and intervention response, as well as potential targets for intervention. Developing improved clinical tools to quantify neural risk factors, as opposed to relying on clinical judgment alone, may improve the detection of SRD risk, particularly if integrated with genetic risk. A recent study demonstrated that brain activation mediated the relationship between a polymorphism in the gene *BDNF* and reading [80]. Furthermore, neural activation may be an active target to improve reading performance for individuals who do not respond to intervention, using techniques such as transcranial direct current stimulation (tDCS). For example, tDCS applied to the left posterior temporal cortex resulted in an increase in word reading efficiency in adult poor readers [81]. In children, tDCS applied to the temporo-parietal regions led to long-lasting improvements in reading 6 months afterwards [82]. Across studies, these effects were particularly effective in poor readers, with specific improvements in word decoding for adults and non-word and low-frequency word reading in younger children [83], although there have been mixed results depending on the parameters used [84]. Other methodologies, including transcutaneous auricular vagus nerve stimulation, also led to improvements in decoding and automaticity of learning novel letter–sound relationships in a new language [85]. This literature, while still developing, is also a promising clinical application for imaging genetics research, particularly for individuals who may be biologically identified as being at risk or unlikely to respond to intervention.

While this study helped demonstrate the clinical relevance of research literature, findings also identified targets of future research. In larger samples used in future research, it would also be helpful to increase the use of other measures, such as functional connectivity measures that give insight into the reading network as a whole. Examining the relationships between multiple structures using asymmetry may provide additional insight into neural factors related to SRD. Using larger samples, particularly in a longitudinal study, to examine asymmetry, as well as novel techniques such as examining the shape, would also allow more statistical analyses that may help to better understand the causality and relationship of these phenotypes to specific reading skills. Analyses of lateralization and asymmetry may provide further insight into the development of reading and could serve as measures of progress in reading or predisposition to reading difficulties. Furthermore, the specific genes and brain structures examined in the current study are promising targets for future research. In terms of translational research, it is also important to investigate clinical impact in diverse populations, as allele frequencies and genetic risk may differ among individuals with different backgrounds [2]. As these future research directions are tackled, they will provide increased knowledge about the biological etiology of SRD and how these biological risk factors interact with the environment, improving our ability to understand interacting risk factors and contribute to early diagnosis and intervention. Ultimately, sample and population studies of brain and genetic risk factors for various developmental disorders are meant to result in findings that are usable in translational applications and utilizable for the purposes of precision diagnostics and treatment. This is an attempt to make a step forward toward such translational applications.

## Figures and Tables

**Figure 1 jpm-13-00156-f001:**
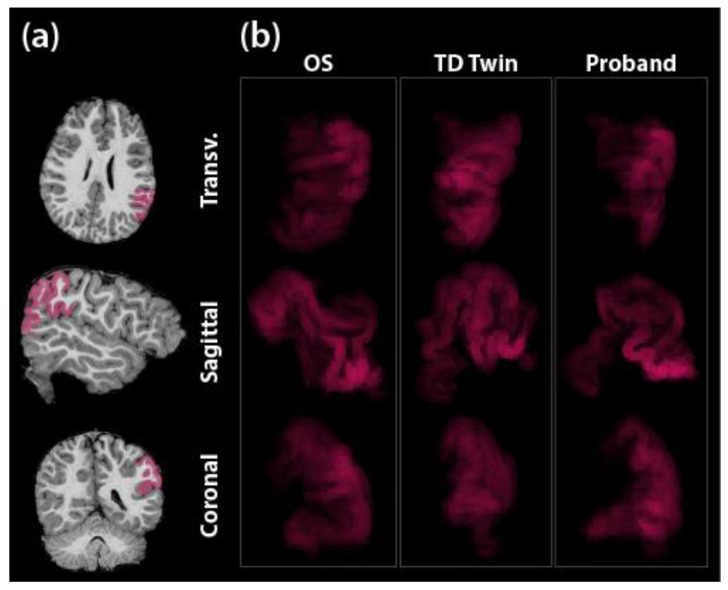
Composite images constructed from the MRI data for the supramarginal gyrus. (**a**) Masked areas are superimposed over anatomy in Freesurfer, on left. (**b**) Composite images constructed from layering many such masks are shown on the right. Each column shows the composite images of one sibling: the older sibling (OS) and the twins (TD Twin and Proband), respectively; rows represent the three anatomic axes. Images are semi-transparent such that brighter areas indicate higher volume areas—or more precisely, a brighter pixel is one that was present in a greater number of mask slices. Colorization is arbitrary and is used in masking to visually differentiate the mask from the underlying grayscale MRI image; grayscale versions were used in analysis.

**Figure 2 jpm-13-00156-f002:**
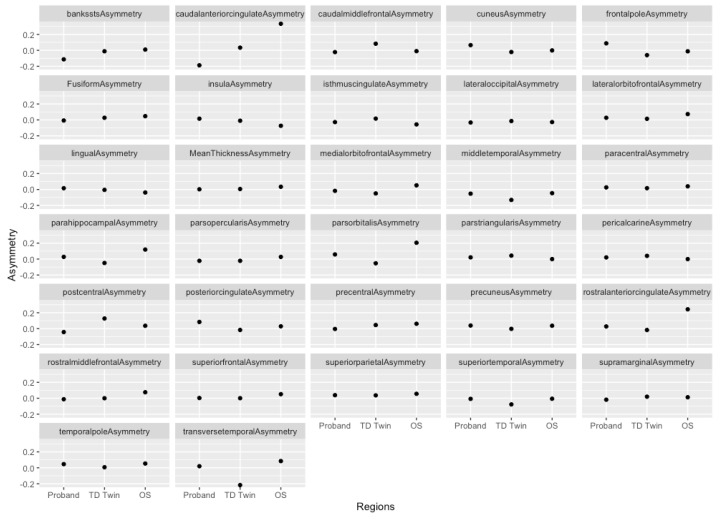
Plots of cortical thickness asymmetry for each sibling.

**Figure 3 jpm-13-00156-f003:**
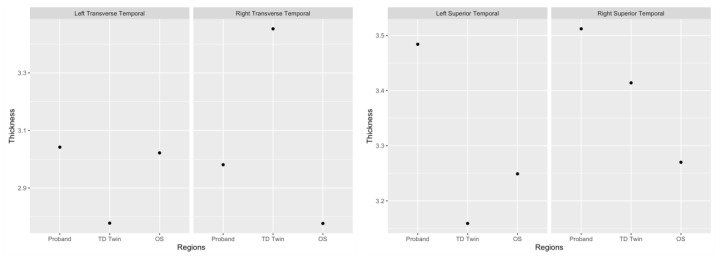
Plots of cortical thickness in the transverse temporal gyrus and superior temporal gyrus.

**Figure 4 jpm-13-00156-f004:**
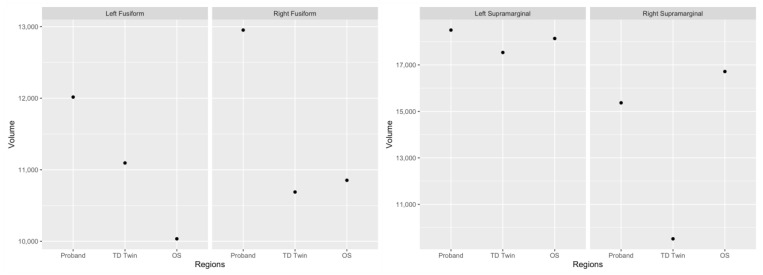
Plots of grey matter volume in the fusiform gyrus and supramarginal gyrus.

**Figure 5 jpm-13-00156-f005:**

Plots of cortical surface area in the fusiform gyrus, supramarginal gyrus, and transverse temporal gyrus.

**Figure 6 jpm-13-00156-f006:**
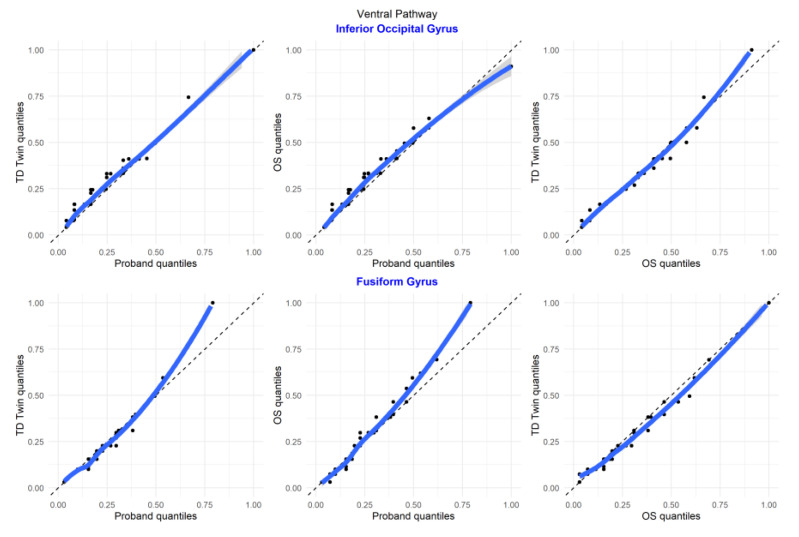
Q–Q plots of sibling comparisons of regions in the ventral pathway.

**Figure 7 jpm-13-00156-f007:**
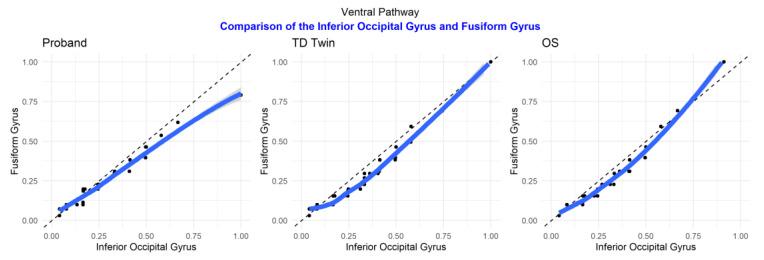
Q–Q plots of region of interest comparisons for regions in the ventral pathway.

**Figure 8 jpm-13-00156-f008:**
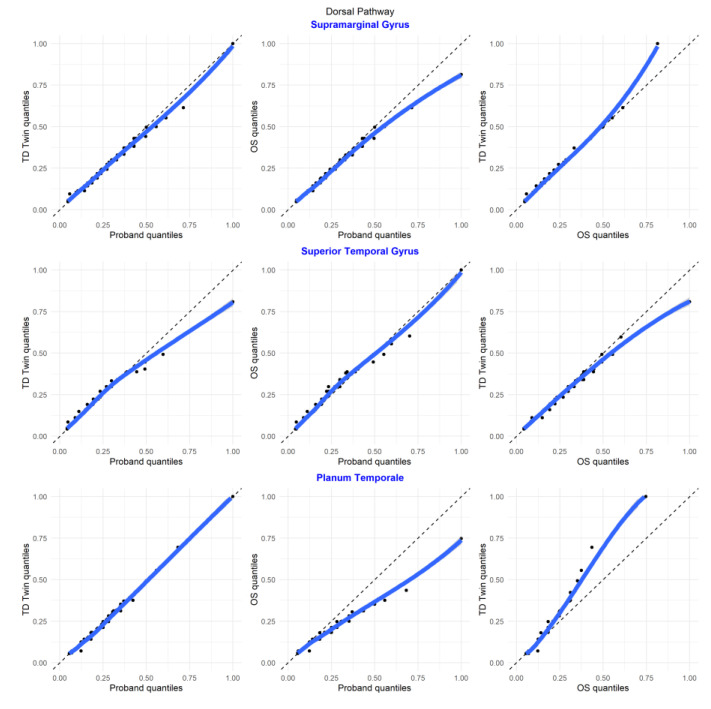
Select Q–Q plots of sibling comparisons of regions in the dorsal pathway.

**Figure 9 jpm-13-00156-f009:**
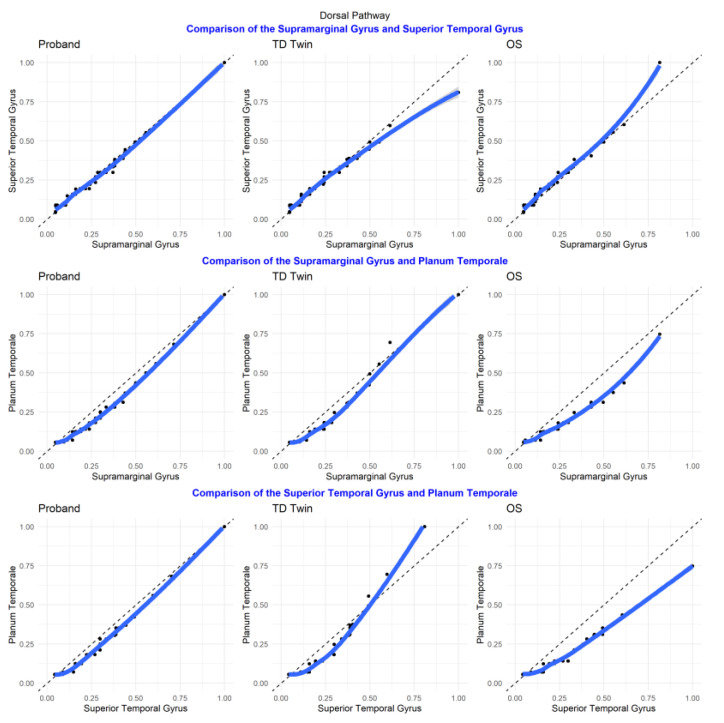
Select Q–Q plots of region of interest comparisons for regions in the dorsal pathway.

**Figure 10 jpm-13-00156-f010:**
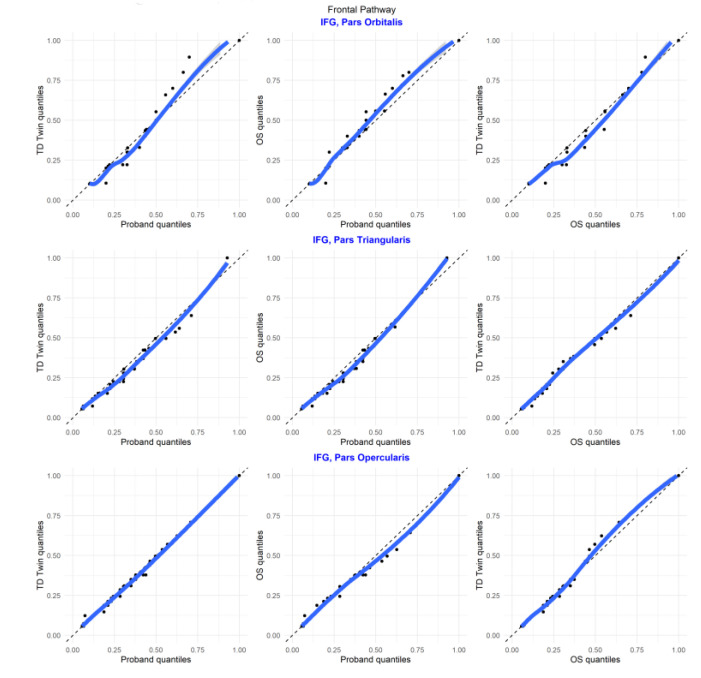
Q–Q plots of sibling comparisons of regions in the frontal pathway.

**Figure 11 jpm-13-00156-f011:**
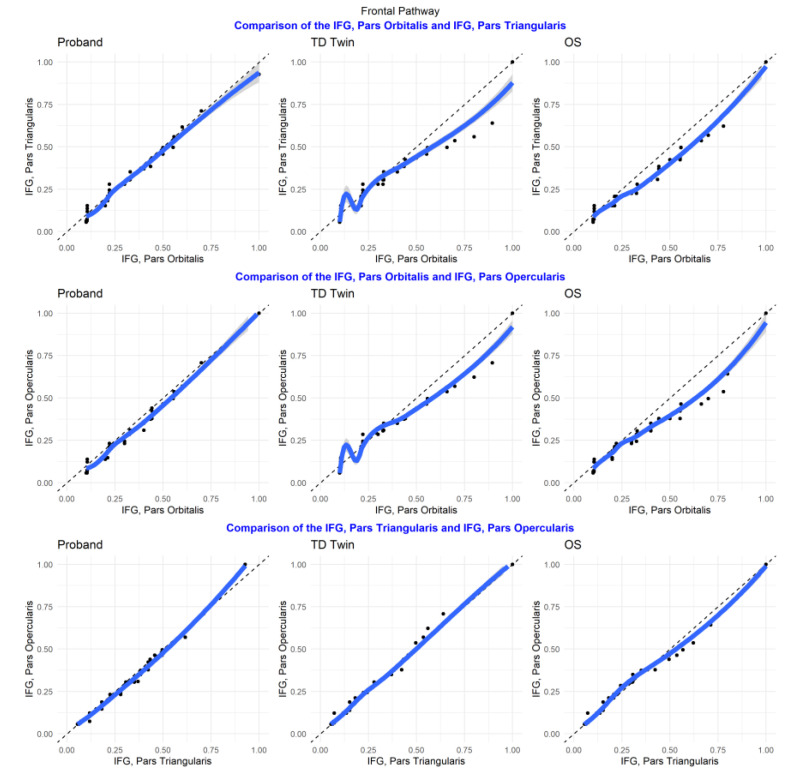
Q–Q plots of region of interest comparisons for regions in the frontal pathway.

**Table 1 jpm-13-00156-t001:** Regions of interest (ROIs) used for imaging analyses.

Pathway Name	Relevant Function	Regions of Interest	*n* Pairs for Image Analysis
Ventral Pathway	Phonological processing	Inferior occipital; Fusiform gyrus	1
Frontal Pathway	Attention and verbalization	Inferior frontal gyrus, pars orbitalis; Inferior frontal gyrus, pars triangularis; Inferior frontal gyrus, pars opercularis	3
Dorsal Pathway	Automatic word recognition	Supramarginal gyrus; Angular gyrus; Superior temporal gyrus; Transverse temporal gyrus; Planum temporale; Planum polare	15

**Table 2 jpm-13-00156-t002:** Sibling behavioral data.

Participant	KTEA Letter/Word Identification	KTEA Nonsense Word Decoding	KTEA Reading Comprehension	KTEA Listening Comprehension
Proband	82	90	82	106
TD Twin	105	113	103	109
OS	90	84	83	93

**Table 3 jpm-13-00156-t003:** Asymmetry values of cortical thickness, cortical surface area, and grey matter volume in reading-related ROIs. Positive values indicate leftward asymmetry and negative values indicate rightward asymmetry.

Phenotype	ID	Fusiform Gyrus	Banks STS	Pars Opercularis	Pars Triangularis	Supramarginal	Transverse Temporal	Superior Temporal
Cortical Thickness	Proband	−0.008	−0.113	−0.022	0.021	−0.018	0.020	−0.008
	TD Twin	0.027	−0.012	−0.022	0.043	0.019	−0.217	−0.078
	OS	0.047	0.009	0.027	−0.001	0.012	0.084	−0.006
Grey Matter Volume	Proband	−0.075	−0.079	0.042	−0.117	0.185	0.126	0.043
	TD Twin	0.037	−0.240	0.083	0.004	0.593	0.247	0.141
	OS	−0.078	0.123	0.051	−0.136	0.082	0.359	0.117
Cortical Surface Area	Proband	−0.084	−0.007	0.072	−0.121	0.225	0.203	0.024
	TD Twin	−0.016	−0.213	0.113	0.015	0.551	0.435	0.203
	OS	−0.113	0.077	0.076	−0.105	0.054	0.320	0.127

**Table 4 jpm-13-00156-t004:** Summary of primary results.

Genetics Results	
Identified Genes	10% of SNPs fit with the expected pattern: 33 SNPs identified in *ZNF385D*, 17 SNPs in *LPHN3*, 9 SNPs in *CNTNAP2*, 2 SNPs in *FGF18*, 2 SNPs in *NOP9*, 11 SNPs in *CMIP*, 4 SNPs in *MYO18B*, and 1 SNP in *RBFOX2*
**Asymmetry Analyses**	
Cortical Thickness	ICC findings demonstrated that cortical thickness of reading-related regions was more similar for the siblings with SRD, followed by the twins, then the TD twin and OS pair. In the transverse temporal gyrus and superior temporal gyrus, the siblings with SRD demonstrated more symmetric or leftward asymmetry while the TD twin had a greater rightward asymmetry of cortical thickness. The greater rightward asymmetry of the TD twin may reflect compensation, as cortical thickness tends to be impacted by experience.
Grey Matter Volume	ICC findings demonstrated that GMV was similar across each sibling pair. In the fusiform gyrus and supramarginal gyrus, the TD sibling had greater leftward asymmetry of GMV, indicating greater volume in the left hemisphere. The reduced leftward asymmetry of GMV in the siblings with SRD may reflect genetic vulnerability.
Cortical Surface Area	ICC findings demonstrated similar correspondence for cortical surface area between all sibling pairs for reading-related regions of interest. The TD twin tended to have more leftward asymmetry than the siblings with SRD, indicating greater cortical surface area in the left hemisphere as compared to right. The reduced leftward asymmetry of surface area in the siblings with SRD may reflect genetic vulnerability, as cortical surface area tends to be impacted by genetic risk.
**Image Analyses**	
Ventral Pathway	In the fusiform gyrus, the TD twin and older sibling were most similar in luminosity distributions. In the inferior occipital region, the twins were more similar than the other sibling pairs in the distribution of luminosity. The TD twin and older sibling may be more similar in luminosity distributions as reflective of current reading ability, as their reading levels are likely comparable.
Dorsal Pathway	In the dorsal pathway, most regions were more similar between the twins (supramarginal gyrus, planum temporale, transverse temporal gyrus) compared to the other sibling pairs. For the superior temporal region, the siblings with SRD were more similar, which may reflect shared vulnerabilities in phonological processing.
Frontal Regions	Most frontal regions were similar between all sibling pairs.

## Data Availability

Not applicable.

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
