# Peer review of "Exploring Genetic and Neural Risk of Specific Reading Disability within a Nuclear Twin Family Case Study: A Translational Clinical Application"

_jpm, 2023, doi:10.3390/jpm13010156_

Round 1

Reviewer 1 Report

This article explored the genetic and neuroanatomical overlaps between siblings in one family. The proband (a DZ twin) has a specific reading disability as did the older sibling, but the co-twin has typical reading. The authors found that the proband and older sibiling had 10% of reading related SNPs in common. Additionally the proband and older sibling shared neuroantomy features for ROIs within the reading network, specifically cortical thickness, asymmetry, cortical surface area, and luminosity. The authors discuss the relevance of genetic and neuroanatomical findings in comparison to the broader literature. 

Major problems:

In the current form it is hard to summarize the main imaging results. This lack of summary is then problematic for the discussion, which is very broad.

The discussion is more far reaching than necessary. Because this work is exploratory, I would encourage the authors to narrow the scope of their discussion - especially the imaging results. The last 4 paragraphs of the discussion are very useful. I would like to see a similar level of synthesis and clarity in throughout the discussion.

Minor:

The figures are hard to read. The resolution is poor because of faceting over so many variables. I suggest moving low interest figures to a supplemental and focusing on high quality relevant figures for the main text. See figure 9 for example of this problem.

For figures provide more detailed/informative titles instead of variable names. 

Please use different ids for children - eg. Proband, TD Twin, OS - in figures and tables.

Some sort of summary table of the results would be helpful in tracking all the different findings. 

Subheaders in the discussion - eg. overlapping genetics; imagining results related to specific reading disability - would improve clarity and focus of discussion. 

Author Response

We appreciate your close reading of the manuscript. Your corrections have been addressed, with our adjustment below included in italic font. Changes in the manuscript have been indicated in track changes.

Comments and Suggestions for Authors

This article explored the genetic and neuroanatomical overlaps between siblings in one family. The proband (a DZ twin) has a specific reading disability as did the older sibling, but the co-twin has typical reading. The authors found that the proband and older sibiling had 10% of reading related SNPs in common. Additionally the proband and older sibling shared neuroantomy features for ROIs within the reading network, specifically cortical thickness, asymmetry, cortical surface area, and luminosity. The authors discuss the relevance of genetic and neuroanatomical findings in comparison to the broader literature. 

Major problems:

In the current form it is hard to summarize the main imaging results. This lack of summary is then problematic for the discussion, which is very broad.

A table to condense and summarize the main imaging results was created to narrow down the most relevant results.

The discussion is more far reaching than necessary. Because this work is exploratory, I would encourage the authors to narrow the scope of their discussion - especially the imaging results. The last 4 paragraphs of the discussion are very useful. I would like to see a similar level of synthesis and clarity in throughout the discussion.

Some of the fine-grained analyses of the results were eliminated, and the discussion/results were better synthesized and clarified.

Minor:

The figures are hard to read. The resolution is poor because of faceting over so many variables. I suggest moving low interest figures to a supplemental and focusing on high quality relevant figures for the main text. See figure 9 for example of this problem.

Figures were condensed to include only relevant figures. Other figures were moved to supplementary material.

For figures provide more detailed/informative titles instead of variable names. 

Figures were edited to provide more informative titles (e.g., names of ROIs rather than variable names).

Please use different ids for children - eg. Proband, TD Twin, OS - in figures and tables.

IDs were changed to Proband, TD Twin, and OS for all tables and figures.

Some sort of summary table of the results would be helpful in tracking all the different findings. 

A  table with a summary of primary results was added to the discussion section.

Subheaders in the discussion - eg. overlapping genetics; imagining results related to specific reading disability - would improve clarity and focus of discussion. 

Subheaders were added to the discussion question for clarity.

Reviewer 2 Report

The title does not really explain the study

Put in: specific reading disability (SRD) in title rather than just SRD.  The word twin should also be in the title as it more than just a family case study.

Suggest for consideration:

Exploring Genetic and Neural Risk Factors of Specific Reading Disability (SRD) using a Twin, Family Case Study Design: A Clinical Application.

Over the paper reports a well conducted study and it worthy of publication. The authors need however, to be more mindful of the readership for a paper on reading difficulties. The reader it not expected to be an expert on DNA and medical imagery terminology and so to improve the comprehension and so value of the research the authors need to better explain their terminology and use of abbreviations. The first time an abbreviation is used the full name needs to be provided then the abbreviation. Even common abbreviations when first reported need to be in full such as ASD and ADHD (line 160).

It is assumed the abbreviation ROI refers to region of interest. This needs to be stated to enhance the comprehension of the paper.

In the abstract and across the paper the abbreviation SNP is dropped in but not explained. Given its important to the paper it has to defined and explained. For example, Single Nucleotide Polymorphisms (SNPs) are the most common type of genetic variation among people and are found in the DNA between genes, and so act as biological markers.

In Figure 1 it is hard to read the title on the images. Could add information in title below such as: image set 1 sibling; image set 2 TD; image set 3 Twin SRD.

Line 317 Behavioral data (see Table 2) revealed low average reading abilities for one twin and the older sibling, and average reading abilities for the second twin. Is the below average based on the aged norms of the test, if so need to state, below average to what?

Table 2 need to keep the same terms across the paper Twin 1 (SRD) Twin 2 (TD) not Twin 1 and 2

As noted in Table 2 the result for the older sibling suggest that this child also has reading difficulties with her listening comprehension lower than the twin with SDR and a similar level with the twin SDR in reading comprehension.

If so, the sibling’s image and DNA profile should be more similar to the twin with SDR than the twin TD without reading problems. Looking at Table 3 unsure if this is the case, although there is a developmental (age) difference that may need to be considered. This could be explained more.

Table 3, it is assumed TWOO1 is twin with SRD, TWOO2 is twin TD and TWOO3 is older sibling. Need to have a key if changing labels.

The plots are interesting and from my reading, there are two factors in play, one age and maturation and the other a variation that influences reading. If SRD and TD are similar but different to Sib there is a development (maturation of the network) suggested. If there is similarity in plots between SRD and Sib, but not TD then this plot is more influenced by reading difficulty. If all three plots are similar, then this variable does not discriminate between the children on reading difficulty or short-term maturation. Thus, based on plot analysis the SRD pattern is suggested more with: (row 4 column 1); (row 5 column I) (row 1 column 3) and (row 7 column 2). By having the older sibling with SRD, the authors have indirectly controlled for short term maturation of the neuro network. Thus, making this a more unique study as maturation of the neuro network is associated with changes in neuro patterns and learning.

In the information under Figure 2, the authors need to direct the reader to the evidence.

Lines 353-55: The TD sibling appeared to have lower cortical thickness in the left hemisphere as compared to the SRD siblings, (see XXX) and higher cortical thickness in the right hemispheric transverse temporal gyrus as compared to the SRD siblings (see XXX).

The section on 3.5.1. Ventral pathway is highly technical, but the authors need to explain why Ventral pathways are of interest and relevance to this study and what is the expected plots. Similarly, the Dorsal pathways information also needs to be explained, why investigate it and its relevance. The fact the different pairs are reported does not explain what the authors are investigating. The QQ plot lines are also not well explained so the reader has to assume what each line represents. One of the problems with just downloading and importing a generated image is assuming it can be interpreted by someone not involved in its generation.

One of the problems is, many of the pathway images are so small that the reader has difficult interpreting them. The authors need to direct the reader to which are important and why.

The discussion section is sound and provides an overview of the results and links the findings to the relevant literature.

The authors have not identified the limitations of their research and this needs to be included. Reading difficulties is a complex domain and there are a significant number of different variables that influence reading development over time. Even if a child has a “genetic” profile that is associated with a reading problem it does not mean it is inevitable, unavoidable or it is predetermined and preordained. Similarly, if the child does not have a particular “genetic” profile that the child will not have a reading or learning problem.  Genetic and neural risk researchers need to be careful to frame their research findings, so they are not indirectly suggesting a return to the “dark days of Eugenics research”.

Overall, the paper adds to knowledge and is worthy of publication, but it needs minor modification to enhance its comprehension, readability and interpretation.

Author Response

We appreciate your close reading of the manuscript. Your corrections have been addressed, with our adjustment below included in italic font. Changes in the manuscript have been indicated in track changes.

Comments and Suggestions for Authors

The title does not really explain the study

Put in: specific reading disability (SRD) in title rather than just SRD.  The word twin should also be in the title as it more than just a family case study.

Suggest for consideration:

Exploring Genetic and Neural Risk Factors of Specific Reading Disability (SRD) using a Twin, Family Case Study Design: A Clinical Application.

The title was changed to “Exploring Genetic and Neural Risk of Specific Reading Disability within a Nuclear Twin Family Case Study: A Translational Clinical Application”

Over the paper reports a well conducted study and it worthy of publication. The authors need however, to be more mindful of the readership for a paper on reading difficulties. The reader it not expected to be an expert on DNA and medical imagery terminology and so to improve the comprehension and so value of the research the authors need to better explain their terminology and use of abbreviations. The first time an abbreviation is used the full name needs to be provided then the abbreviation. Even common abbreviations when first reported need to be in full such as ASD and ADHD (line 160).

We appreciate the reviewer’s comments that the study is paper of publication. Full names of the abbreviations for ASD and ADHD were included (line 80, line 160).

It is assumed the abbreviation ROI refers to region of interest. This needs to be stated to enhance the comprehension of the paper.

The full name for ROI was included (line 149)

In the abstract and across the paper the abbreviation SNP is dropped in but not explained. Given its important to the paper it has to defined and explained. For example, Single Nucleotide Polymorphisms (SNPs) are the most common type of genetic variation among people and are found in the DNA between genes, and so act as biological markers.

An explanation of SNPs was added (lines 72-75).

  • “Specifically, much of the genetic research on SRD has been focused on single nucleotide polymorphisms (SNPs), or single substitutions of nucleotides in a genome sequence, which can serve as biological markers or predict genetic risk.”

In Figure 1 it is hard to read the title on the images. Could add information in title below such as: image set 1 sibling; image set 2 TD; image set 3 Twin SRD.

Figure 1 was edited to make titles clearer, with more explanation added in the description as well.

Line 317 Behavioral data (see Table 2) revealed low average reading abilities for one twin and the older sibling, and average reading abilities for the second twin. Is the below average based on the aged norms of the test, if so need to state, below average to what?

Clarification was added that the low average was “based on age norms of the tests.” (Line 326).

Table 2 need to keep the same terms across the paper Twin 1 (SRD) Twin 2 (TD) not Twin 1 and 2

Labels were altered in Table 2 for consistency (to Proband, TD Twin, and Older Sibling (OS).

As noted in Table 2 the result for the older sibling suggest that this child also has reading difficulties with her listening comprehension lower than the twin with SDR and a similar level with the twin SDR in reading comprehension. If so, the sibling’s image and DNA profile should be more similar to the twin with SDR than the twin TD without reading problems. Looking at Table 3 unsure if this is the case, although there is a developmental (age) difference that may need to be considered. This could be explained more.

This pattern is described throughout the results and the discussion.

Table 3, it is assumed TWOO1 is twin with SRD, TWOO2 is twin TD and TWOO3 is older sibling. Need to have a key if changing labels.

Labels were updated in Table 3 and Figure 2.

The plots are interesting and from my reading, there are two factors in play, one age and maturation and the other a variation that influences reading. If SRD and TD are similar but different to Sib there is a development (maturation of the network) suggested. If there is similarity in plots between SRD and Sib, but not TD then this plot is more influenced by reading difficulty. If all three plots are similar, then this variable does not discriminate between the children on reading difficulty or short-term maturation. Thus, based on plot analysis the SRD pattern is suggested more with: (row 4 column 1); (row 5 column I) (row 1 column 3) and (row 7 column 2). By having the older sibling with SRD, the authors have indirectly controlled for short term maturation of the neuro network. Thus, making this a more unique study as maturation of the neuro network is associated with changes in neuro patterns and learning.

We appreciate your analysis and comments.

In the information under Figure 2, the authors need to direct the reader to the evidence.

Lines 353-55: The TD sibling appeared to have lower cortical thickness in the left hemisphere as compared to the SRD siblings, (see XXX) and higher cortical thickness in the right hemispheric transverse temporal gyrus as compared to the SRD siblings (see XXX).

Reference to figure 3 was clarified.

The section on 3.5.1. Ventral pathway is highly technical, but the authors need to explain why Ventral pathways are of interest and relevance to this study and what is the expected plots. Similarly, the Dorsal pathways information also needs to be explained, why investigate it and its relevance.

Clarification on the roles of the ventral pathway (“related to automatic recognition of words in reading”) and dorsal pathway (“involved in phonological processing in reading”) were added as a reminder to the reader.

The fact the different pairs are reported does not explain what the authors are investigating. The QQ plot lines are also not well explained so the reader has to assume what each line represents. One of the problems with just downloading and importing a generated image is assuming it can be interpreted by someone not involved in its generation.

Additional explanation was added when introducing the QQ plot results as a refresher to the reader in section 3.5.1: “QQ plots were observed to better understand the similarity between luminosity, reflective of shared structure, between each sibling pair (labeled on the X and Y axes) for each brain ROI.”

One of the problems is, many of the pathway images are so small that the reader has difficult interpreting them. The authors need to direct the reader to which are important and why.

The labels on the images have been clarified so that it is clear which graphs are being referred to in the text. Furthermore, images were condensed to only the most important figures.

The discussion section is sound and provides an overview of the results and links the findings to the relevant literature.

We thank the reviewer for their positive feedback.

The authors have not identified the limitations of their research and this needs to be included. Reading difficulties is a complex domain and there are a significant number of different variables that influence reading development over time. Even if a child has a “genetic” profile that is associated with a reading problem it does not mean it is inevitable, unavoidable or it is predetermined and preordained. Similarly, if the child does not have a particular “genetic” profile that the child will not have a reading or learning problem.  Genetic and neural risk researchers need to be careful to frame their research findings, so they are not indirectly suggesting a return to the “dark days of Eugenics research”.

These limitations were discussed in the discussion section (line 747): “Limitations of the current methodologies include the inherent difficulties of conducting a case study with a limited number of participants; while descriptive examinations of relevant variables could be studied, it is impossible to determine causality without a larger sample to conduct direct analyses. While these studies demonstrate the link between asymmetry of brain structures and reading, it is difficult to determine directionality or causality of these effects, and whether asymmetry influences reading or vice versa, or whether this interaction may change over time (Bishop et al., 2013). Furthermore, SRD is complex and can be impacted by a multitude of factors in addition to neural and genetic risk factors. Even children who may be at genetic risk for developing SRD do not always develop SRD. Similarly, even if a child is not at genetic risk, they may develop SRD due to other risk factors. Understanding more about biological risk factors simply serves as a starting point to identify children who may benefit from further monitoring or intervention.

Overall, the paper adds to knowledge and is worthy of publication, but it needs minor modification to enhance its comprehension, readability and interpretation.

Thank you for your comments.